# Using Nutritional Strategies to Shape the Gastro-Intestinal Tracts of Suckling and Weaned Piglets

**DOI:** 10.3390/ani11020402

**Published:** 2021-02-05

**Authors:** Anne M.S. Huting, Anouschka Middelkoop, Xiaonan Guan, Francesc Molist

**Affiliations:** Research & Development, Schothorst Feed Research B.V., 8218 NA Lelystad, The Netherlands; AHuting@schothorst.nl (A.M.S.H.); AMiddelkoop@schothorst.nl (A.M.); XGuan@schothorst.nl (X.G.)

**Keywords:** creep feed, early life, feed intake, gut function, health, nutrition, pig, pre-weaning, post-weaning, supplemental milk

## Abstract

**Simple Summary:**

Throughout the world, piglet mortality and morbidity in large litters are a major welfare concern and source of economic losses. Gastro-intestinal problems rank amongst the highest causes of morbidity, mortality and antimicrobial use. As evidenced in the recent literature, nutritional interventions before and after weaning can modulate gut development, thereby reducing the risk of gastro-intestinal problems. In particular, early-life nutrition has begun to receive increasing interest, given its potential to modulate gut health in the long-term. The literature nevertheless contains little information on how pre-weaning and post-weaning nutritional strategies can be combined to sustain optimal gut health throughout the challenging process of weaning. To address this gap in current knowledge, this review summarises a large body of literature on nutritional strategies aimed at supporting gut health in piglets, combining individual strategies into a structured nutritional approach over time, starting from a few days after birth to 5–6 weeks post-weaning. The review also contains propositions concerning potential avenues for future research that may contribute to the reduction in gastro-intestinal problems and the associated use of antimicrobials in the pig industry.

**Abstract:**

This is a comprehensive review on the use of nutritional strategies to shape the functioning of the gastro-intestinal tract in suckling and weaned piglets. The progressive development of a piglet’s gut and the associated microbiota and immune system offers a unique window of opportunity for supporting gut health through dietary modulation. This is particularly relevant for large litters, for which sow colostrum and milk are insufficient. The authors have therefore proposed the use of supplemental milk and creep feed with a dual purpose. In addition to providing nutrients to piglets, supplemental milk can also serve as a gut modulator in early life by incorporating functional ingredients with potential long-term benefits. To prepare piglets for weaning, it is important to stimulate the intake of solid feed before weaning, in addition to stimulating the number of piglets eating. The use of functional ingredients in creep feed and a transition diet around the time of weaning helps to habituate piglets to solid feed in general, while also preparing the gut for the digestion and fermentation of specific ingredients. In the first days after weaning (i.e., the acute phase), it is important to maintain high levels of feed intake and focus on nutritional strategies that support good gastric (barrier) function and that avoid overloading the impaired digestion and fermentation capacity of the piglets. In the subsequent maturation phase, the ratio of lysine to energy can be increased gradually in order to stimulate piglet growth. This is because the digestive and fermentation capacity of the piglets is more mature at this stage, thus allowing the inclusion of more fermentable fibres. Taken together, the nutritional strategies addressed in this review provide a structured approach to preparing piglets for success during weaning and the period that follows. The implementation of this approach and the insights to be developed through future research can help to achieve some of the most important goals in pig production: reducing piglet mortality, morbidity and antimicrobial use.

## 1. Introduction

With the objective of improving overall efficiency within the swine industry, breeding has traditionally focussed on carcass traits and growth rate, as well as on the number of piglets produced per sow/year (prolificacy). Litter size has thereby increased considerably in recent decades, resulting in complications relating to animal management, health and welfare, as reflected in increased morbidity and prenatal and neonatal mortality [1].

Increases in prolificacy have also led to an increase in the number of piglets with low birth weight (see Figure 1; [1,2]) and piglets that have been subjected to intra-uterine growth retardation (IUGR), which currently affects 30–40% of all piglets [3]. In addition to being at greater risk of pre-weaning mortality [4,5], light-born and IUGR piglets that do survive tend to thrive less efficiently [6,7,8] and to be more susceptible to diseases [1,9,10]. In addition, the genetic selection for leaner meat has resulted in piglets being born with limited body reserves [11], which fail to meet their early-life requirements for maintenance, thermoregulation and activity [12]. At the same time, colostrum yield is independent of litter size, meaning that increases in litter size decrease the amount of colostrum consumed per piglet [13]. On top, IUGR piglets consume less colostrum than piglets with a normal head morphology [14] and also have a lower stomach capacity to do so [15]. This is important, as timely and sufficient colostrum is essential to the survival and lifetime performance of piglets [9,16,17]. In addition, the number of teats has not increased in relation to the number of piglets born alive [18], thereby increasing competition for the already limited resources. Taken together, reductions in the intake of colostrum and milk increase the risk of malnutrition or even starvation, as well as the risk of hypothermia and disease susceptibility, ultimately resulting in variable growth rates within litters [1]. This has introduced new challenges with regard to keeping all piglets alive and healthy throughout production. The progressing development of a piglet’s gut, as well as the associated gut microbiota and immune system during the first weeks of life [19], offers a unique window of opportunity for the early life programming of the gastro-intestinal tract.

The various stressors to which piglets are exposed at weaning in combination with their impaired gut and immune function make newly weaned piglets extremely vulnerable to post-weaning diseases, which consist largely of gastro-intestinal infections [20]. As a result, most antimicrobials (i.e., antibiotics and microminerals) within the context of pig production are used in the nursery barn [21]. Risk factors for post-weaning diseases include weaning age, weaning weight, low feed intake during the immediate post-weaning period, overeating (i.e., large amounts of feed consumed in a low number of meals), large amounts of undigested nutrients arriving at the end of the ileum and several management factors (e.g., hygiene, temperature, draught, feeding spaces) [22,23]. In the past, nutritionists could mask gastro-intestinal infections by using in-feed antibiotics, pharmaceutical levels of zinc (Zn) and/or high levels of copper (Cu) in piglet diets [21]. Since 2006, however, the use of in-feed antibiotics has been prohibited in Europe [24], and further restrictions with respect to Cu and Zn in piglet diets are either already in place or are expected to be implemented in the near future [25,26,27]. Combined with the fact that gastro-intestinal problems rank amongst the highest causes of morbidity and mortality in pig production [28,29,30], these restrictions have created a need for other dietary interventions aimed at improving the health of the gastro-intestinal tract (“gut health”, as discussed by Pluske et al. [31]).

In response to the developments outlined above, this review focusses on nutritional solutions intended to prepare piglets for success during the weaning process and the period thereafter (5–6 weeks post-weaning), based on the latest developments in practical swine nutrition. A varied array of dietary interventions and their relationship to gut health is discussed and combined into nutritional strategies for supporting piglets during early life and the subsequent weaning process. Recent reviews have charted the potential of using nutritional strategies in sows to modulate piglet gut health, particularly at birth [19,32]. The current review, therefore, focusses on nutritional interventions for suckling and weaned piglets. The strategies addressed include oral supplements (first days after birth), supplemental milk (first weeks after birth) and solid feed (pre- and post-weaning). Although several nutritional interventions have been shown to give piglets a good start, not all weaned piglets are alike. This review therefore proposes a structured nutritional approach over time, suggesting potential directions to explore in order to shape the gastro-intestinal tract of all young piglets to reduce morbidity, mortality and antimicrobial use in pig-production systems.

## 2. Early Nutrition Interventions

Litters consisting of more piglets than the number of productive sow teats are likely to require additional management and nutritional interventions to increase pre-weaning survival, as the sow cannot rear them on her own [1]. The various management strategies for rearing piglets in large litters, their effectiveness and their implications for animal welfare have been described in reviews by Baxter et al. [33,34]. This section highlights the effect of dietary components that can improve gut health and development in oral supplements and supplemental milk.

As piglets are born without passive immune protection from the sow, they are dependent on the transmission of immunoglobulins through sow colostrum. Colostrum is also an important source of energy (e.g., for thermoregulation, energy deposition, physical activity and maintenance), which contributes to early survival [16]. In addition, the ingestion of colostrum has been associated with substantial growth in the intestine (cellular proliferation) [35], as well as with major changes in the shape, size and density of villi [36,37]. This suggests that, in addition to its importance for immune function and providing energy, colostrum plays an important role in the development of the gastro-intestinal tract. The increasing trend in the number of piglets per sow has resulted in the birth of less viable piglets and a lower colostrum intake per piglet. It is therefore not surprising that oral supplementation (e.g., with colostrum or energy) during the first 12 h after birth—in order to promote colostrum intake and thus early survival—has gained interest in recent years. To date, however, only a limited number of studies have evaluated the effectivity of such products on survival and growth, and the studies that have been published vary considerably with regard to timing of the supplements, category of piglets (e.g., birth weight), type of product and dosage. In addition, the sow effect on piglet survival is often poorly controlled [34].

Piglet bacterial diversity during early life has been suggested as an important factor in gut physiology and immunity, and it might influence the susceptibility of piglets to enteric infections in later life [19,21]. At birth, the gut microbiota of piglets is shaped by the microbiota of their sow, including the microbiota existing in the sow’s vaginal tract, as well as in faeces, colostrum/milk and on the teats [38,39]. Salivary microbes constitute another potentially influential source that merits further investigation. The microbiome of the sow—and therefore of her piglets—is particularly influenced by the sow’s diet during gestation and lactation. It therefore offers an opportunity to improve piglet health, as addressed by Ferret-Bernard and Le Huërou-Luron [19] and Jiang et al. [32]. The early gut colonisers of neonatal piglets also include microbes from the environment [38,39]. There are thus many avenues for programming the gut during early life that could be beneficial throughout the entire production life of piglets. This review is limited to the nutritional aspects of oral supplements, supplemental milk and creep feed.

### 2.1. Oral Supplementation

Given that feed intake is limited immediately after birth, studies have examined the oral supplementation of neonatal piglets, including prebiotics (e.g., inulin, β-glucans and oligosaccharides) and functional amino acids (e.g., glutamine). Although most of these studies involve the administration of products on a daily basis for periods (i.e., 7 to 28 days) that are possibly longer than practical, these studies indicate that intestinal morphology and function can be modulated in the long term (at least up to two weeks after oral administration; Table 1). Taken together, these results indicate a potential for early-life programming through dietary modulation. The transplantation of faecal microbiota from the sow or adult pigs to new-born piglets also has the potential to prevent or reduce intestinal diseases (e.g., *Escherichia coli (E. coli)*), as it can influence gut microbiota colonisation, intestinal barrier and immune function (reviewed by Canibe et al. [40]). Negative results have also been reported [40], however, indicating that the technique must first be optimised and pass biosecurity and regulatory barriers before it can be used safely and efficiently in pig production.

### 2.2. Supplemental Milk

Bioactive compounds are contained in sow milk, including hormones, growth factors, immune cells, antimicrobial compounds and commensal bacteria, which play a vital role in establishing a healthy and functional gastro-intestinal tract [48,49]. In some cases, however, as with large litters and impairments in the milking ability of the sow (e.g., due to heat stress, sickness), the application of supplemental milk (i.e., formula milk provided to sow-reared piglets) may help to improve pre-weaning survival and performance by increasing weaning weight and lowering variability in body weight within the litter. Studies on the effectivity of this method have yielded inconsistent results [34]. Nonetheless, supplemental milk may especially be relevant for farms that make little or no use of nurse sows or that apply minimal or no cross-fostering of piglets.

Evidence also suggests that supplemental milk could potentially alter gut health and function. For example, de Greeff et al. [50] report that the provision of supplemental milk (age 1–21 days; average litter size 13–14 piglets) enhanced small intestinal growth and cell proliferation, as well as increased concentrations of short-chain fatty acids (SCFA) in the colon (which is the energy source for intestinal cells). These results were accompanied by softer faeces and an improved weaning weight [50]. In addition to enhanced levels of SCFA, another study (Jin et al. [51]) reports that the provision of supplemental milk from 7 to 21 days of age modulated gut microbiota colonisation in the jejunum and colon (e.g., reduced *E. coli* abundance) and altered mRNA expression of jejunal inflammatory cytokines and barrier proteins. As a result, piglets fed with supplemental milk had a lower incidence of pre-weaning and post-weaning diarrhoea [52]. Taken together, these studies suggest that supplemental milk, in addition to sow milk, has the potential to improve piglet gut health in large litters.

Supplemental milk can be formulated in order to serve both a nutritional and a functional role. The composition of formula milk has been shown to influence gut morphology [53,54], digestive function [53,55], mucosal immunology and microbiota in piglets [56]. It is important to note, however, that these studies focus on the provision of formula milk to artificially reared piglets (further defined as milk replacers in the current review), mostly as a model for human infant formula, rather than the use of formula milk as a supplement for sow-reared piglets. Some of these studies indicate that intervention periods of just 3–8 days could be sufficient to affect gut morphology and (immune) function (Table 2).

For example, Le Huërou-Luron et al. [55] suggest that the composition (a variety of sources) and the structure (a variety of stabilisers) of the added fat used can have an impact on the immune system and gut physiology. They demonstrate that the use of both saturated milk fat and vegetable oils that had been stabilised by a mixture of proteins and milk fat globule membrane fractions given from two days of age could modify intestinal physiology, mucosal immunity, microbial composition and protein digestion at 7 days of age [55]. In addition, prebiotics (either alone or in combination) are often added to infant formulas in order to mimic the effect of “human milk oligosaccharides” (HMO). These compounds are believed to modulate gut microbiota, to have several antimicrobial actions and to play a role in immune development. For example, Radlowski [53] studied the effect of a combination of two prebiotics, polydextrose and short-chain fructo-oligosaccharides (scFOS) in milk replacer, with regard to their ability to mimic the bioactive compound HMO. They found evidence that polydextrose and scFOS in the milk replacer provided during the first two weeks of age yielded effects similar to those of sow milk with respect to intestinal function (e.g., similar peptidase activity) [53]. Similarly, Alizadeh et al. [54] studied the effect of the prebiotic galacto-oligosaccharide (GOS) in milk replacer. They observed morphological changes in the duodenum (e.g., enlarged villus height, villus area and villus-to-crypt ratio) after GOS supplementation in the milk replacer of very young piglets (4 days old). These results suggest that the supplementation may improve the utilisation of nutrients. In the same study, GOS also seemed to improve gut functioning (e.g., nutrient digestion, barrier function) and mucosal immune functioning, although this did not become apparent until after a longer period of exposure (age 27 days) [54]. Taken together, these studies emphasise the potential of modulating piglet gut development in early life through formula milk, while clearly highlighting that the effects of functional ingredients in supplemental milk (for sow-reared piglets) rather than milk replacer (for artificially reared piglets) as used in the reported studies warrant further investigation, as their effects may be less pronounced in sow-reared piglets.

When considering the relative success of supplemental milk (and its possible effects on gut health and development), it is important to understand which piglets within the litter and in which period the supplemental milk is consumed. Whereas Baumann et al. [57] report that piglets only seldom ignore supplemental milk, de Greeff et al. [50] observe that 13% of the piglets in their study were consuming supplemental milk at Week 1 after birth, with 51% doing so at Week 2 and 87% at Week 3. Another study suggests that low birth-weight piglets (≤1.25 kg) consumed more supplemental milk than their heavier littermates (1.6 to 2.0 kg; [58]), while other authors have found little evidence to explain variations in the drinking of supplemental milk by piglets of different body-weight classes [57]. In the latter study, piglets that used the milk feeder frequently between Day 2 and Day 6 after birth had gained less weight on Day 6 than piglets that had visited the milk feeder only occasionally [57]. This suggests that variations in the intake of supplemental milk can likely be explained by the intake of sow milk, which is the main energy source during lactation. In addition to within-litter variations, studies have revealed wide variations between litters with regard to supplemental milk intake, ranging from almost 0 to >20 litres per piglet from birth to three weeks of age [59]. Factors influencing supplemental milk intake between litters include room temperature [59] and the milk production of the sow [60].

Although there is no perfect solution, for large litters and in case of poor sow performance, the provision of oral supplements and/or supplemental milk to piglets may be useful in early life to decrease pre-weaning mortality, while improving gut health and performance. Sow milk should remain the most important source of nutrients, however, as supplemental milk (as it is currently applied) does not provide the same nutrients and bioactive compounds as sow milk. In addition, the economic benefits of a strategy—taking into account both costs (e.g., labour, cost of the supplement, cost of the system used) and long-term revenues—should be considered when determining the efficiency of such strategies, which is likely to differ from one pig producer to another.
animals-11-00402-t002_Table 2Table 2Effects of pre-weaning dietary interventions in milk replacer on the gut development of artificially reared piglets. ↓ Significant decrease at *p* < 0.05; ↑ Significant increase at *p* < 0.05; = No significant difference; BW = body weight.ReferenceDietary Intervention(s) Intervention Period (Age)Age at Sampling Effects on Gut Development versus a Control Milk Replacer[53]Polydextrose (2 g/L) + Fructo-oligosaccharides (2 g/L)1–147, 14Lowers digestive enzyme production at both sampling days:↓ Lactase, sucrase, disaccharidase and aminopeptidase N activity (ileum)↓ Dipeptidyl peptidase IV activity (jejunum, ileum)= Small-intestine weight and length= Cytokine expression (ileum)= Faecal consistency[61]Polydextrose (2 g/L) + Galacto-oligosaccharides (2 g/L)2–3333= Small-intestine weight and length↓ Total volatile fatty acids (colon, no differences in caecum and faeces)↑ Dry matter of colon digesta↓ Dry matter of faeces[53]Galacto-oligosaccharides (2 g/L) + Inulin (2 g/L)1–2121= Small-intestine weight and length= Immune cell population= Nitric oxide synthase activity= Cytokines IL-6 and TNF-α in blood[54]Galacto-oligosaccharides: 0.8%1–264, 27Modulates gut microbial population and improves intestinal morphology and barrier function:↑ Duodenum villus height, villus area, villus-to-crypt ratio (d4), villus width (d27)↑ Jejunum villus height (d27)↓ Caecum pH (d4, no other pH differences) ↑ Caecum butyric acid (d27)↑ Lactobacillus, bifidobacterium in faeces (d27)↑ Maltase activity (colon: d4, caecum: d27)↓ Lactase, sucrase, maltase activity (ileum, d27)↑ mRNA expression of defensins (colon, d4)↑ mRNA expression of tight-junction proteins in different segments of the intestine (d4, d27)↑ Tight-junction protein expression (duodenum, colon, d27)↑ Secretory IgA from saliva (from d19)[62]Galacto-oligosaccharides: 8 mg/mL3–2525↑ Relative large-intestine weight in males, but not females= Small-intestine weight↓ Haematology and clinical chemistry blood parameters[63]Galacto-oligosaccharides (7 g/L) + milk fat globule membrane-10 (5 g/L) + polydextrose (2.4 g/L) + bovine lactoferrin (0.6 g/L)3–3333Modulates gut microbiota population and stimulates gut function:= Intestinal weight and length, small-intestinal morphology↓ Colon area↑ Parabacteroides, Clostridium IV, Lutispora (colon)↓ Mogibacterium, Collinsella, Klebsiella, Escherichia-Shigella, Eubacterium, Roseburia (colon)↑ Lactase activity (jejunum), lactase-to-sucrase activity (duodenum, ileum; no difference in jejunum)= Sucrose activity (in any part)↑ Vasoactive intestinal peptide expression (ileum)[56]Bovine colostrum23–3131Reduces colonisation by *E. coli* and modulates intestinal immune system:↓ Diarrhoea frequency↓ *E. coli* in jejunal and ileal tissue and content in non-inoculated intestinal samples, but not in ETEC F18-inoculated samples↓ TLR-4 and IL-2 gene expression (jejunal and ileal mucosa)= Ig concentrations in mucosa and plasma[55]VP: palm + rapeseed oilVM: VP + milk fat globule membranes (MFGM)MM: VM + sunflower + milk fat2–287, 28VM and MM: ↑ relative jejunum weight and mucosal density at d7 and d28, ↑ ileum mucosal density at d28, = digestive enzyme activity, = epithelial barrier permeabilityMM: ↓ protein digestion (casein),↑ interferon γ secretion from mesenteric lymph node cells, x= bacterial diversity, ↑ Parabacteroides, Escherichia/Shigella, Klebsiella, ↓ Clostridiales Family XIII, Veillonellaceae[64]Fat origin: only plant lipids or a half-half mixture of plant and dairy lipids2–2833= Short-chain fatty acids (faeces)[65]Yeast β-glucans: 5, 50 or 250 mg/L2–217, 21= Small-intestine length and weight= Ileal crypt depth, villus height, ascending colon-cuff depth= T cell phenotypes, cytokine gene expression, cell proliferation[66]Wheat: up to 40%11–25 ^1^25↓ Relative full stomach weight↑ Sucrase, maltase↑ Leukocytes, neutrophils[67]Pectin: 2 g/L or 10 g/L2–23
10 g/L pectin: ↓ feed efficiency, ↓ apparent ileal and total tract digestibility of dry matter, crude protein and energy2 g/L pectin: ↓ apparent total tract digestibility of dry matter^1^ A basal milk replacer was given from Day 3 to Day 11 after birth.

## 3. Preparing Piglets for Weaning

The dietary transition from mainly sow’s milk to exclusively solid feed is one of the important stressors that occur at weaning within the field of commercial pig husbandry [68,69], thus contributing to the post-weaning dip in piglet health and performance. The aim of providing creep feed is to improve post-weaning piglet performance by habituating piglets to solid feed prior to weaning, while often having only marginal effects on pre-weaning production parameters. In a recent review on the effects of creep feed on gut development [70], the author concludes that the effects are mostly ambiguous, particularly with regard to the effects of creep feed on the incidence of diarrhoea [71,72,73]. While most studies report no effects of the provision or intake (eaters vs. non-eaters) of creep feed on gut development [72,73,74], a few studies have reported subtle beneficial effects on gut morphology, cell proliferation and net absorption in the small intestine, most of which were observed post-weaning [75,76,77,78]. As is the case with supplemental milk, creep feed could plausibly be expected to affect gut microbiota colonisation [79,80,81,82,83]. It should be noted, however, that the effects of creep feed on gut (microbiota) development are likely to be dependent on age at provision [84], dietary and nutrient composition [85], and intake level. The importance of intake levels is evident in the fact that the effects of creep feed on piglet performance are pronounced primarily in piglets that consume the creep feed (i.e., eaters), and particularly in those that consume relatively large amounts (reviewed by Middelkoop; [70]). These observations correspond to the findings of Choudhury et al. [82], who report a correlation between the amount of time that piglets spent with their heads in the trough and the change in observed microbiota colonisation. As the amount and dietary composition of feed consumed seem largely responsible for success in influencing gut development, these aspects are discussed in greater detail below.

### 3.1. Pre-Weaning Nutritional Strategies to Stimulate Individual Dry-Matter Intake

The number of piglets consuming creep feed increases exponentially with age [86,87]. Although no comparison has been made within a single experiment, the number of piglets consuming creep feed appears to be lower than the number of piglets drinking supplemental milk, particularly during the first two weeks of lactation (~5% creep-feed eaters [75,86,87] vs. 51% supplemental-milk drinkers [50]). With regard to creep-feed intake, there is evidence that the intake of sow milk plays a role in determining the extent to which piglets are attracted to creep feed [88,89]. In general, posterior teats produce lower quantities [90] and quality [91] of milk than do anterior and middle teats, while teats in the middle part of the udder are more prone to teat disputes than are the others [92]. More specifically, piglets suckling the middle and posterior teats have been observed to consume more creep feed than piglets suckling the anterior teats [86,93]. Birth weight has also been shown to affect the intake of creep feed, although the published correlations with creep-feed intake have been both positive [94,95] and negative [96,97]. This suggests that the birth-weight effects observed might have been confounded by other factors (e.g., teat order and weight distribution within the litter) [86]. Strategies for increasing creep-feed intake and the number of piglets eating range from housing strategies (e.g., intermittent suckling and multi-litter housing) and management strategies (e.g., feeder type and enrichment materials) to nutritional strategies [70]. The nutritional strategies are discussed below.

While piglets have the ability to suckle and drink from birth, their (pre)molars, oral motor skills and mastication muscles must develop over time in order to handle, chew and ingest solid feed [98,99]. To facilitate this process from suckling/drinking (liquid feed) to eating (solid feed), creep feed can be mixed with water or supplemental milk, starting with a relatively high liquid content, and gradually decreasing it over time. In addition to observing that piglets had a higher intake of porridge (creep feed pellets-to-water ratio of 1:3) as compared to pellets in the first three weeks of lactation, Clouard et al. [100] report a higher intake of the pelleted transition diet that was given in the fourth week of lactation.

Softer pellets may also facilitate the transition to solid diets. They can be produced either by adapting the production process (e.g., decreasing die thickness on the pellet press) or changing the diet composition. The most commonly studied method of softening pellets involves increasing pellet diameter. Larger (5–12.7 mm), and thereby softer, pellets have consistently been shown to increase the feed intake of suckling piglets, as compared to conventionally sized pellets of 2–4 mm in diameter, in either early or late lactation (e.g., [101,102,103]). In addition to improving feed intake, a larger pellet diameter might also have beneficial effects on the gut, including reducing the proportion of medicated piglets (mostly for ill thrift) post-weaning when larger pellets (9 vs. 4 mm) were fed pre-weaning [103] and improving post-weaning feed-conversion ratio when larger pellets (5.2 vs. 4 mm) were fed post-weaning [104]. The percentage of eaters pre-weaning is addressed in only one of the studies, with no differences found between pellet diameters of 3.2 and 12.7 mm [102]. In addition, Chen et al. [105] evaluate the effect of pellet hardness on feed intake by testing a hard-pellet creep feed (i.e., hardness of 2690 g) versus a soft-pellet creep feed (i.e., hardness of 505 g). According to their data, piglets receiving the soft pellets from Day 14 to Day 22 after birth had higher creep-feed intake pre-weaning than did those in the other treatment groups. The authors speculate that this might have been a result of the higher moisture content and greater concentration of starch gelatinisation in the softer pellets, which seemed to have a positive influence on the palatability of the feed [105]. The processing of specific feedstuffs to modulate starch gelatinisation rather than the processing of the complete diet might also be useful for stimulating pre-weaning feed intake, as has been observed post-weaning [106].

The composition of the creep feed itself, with a focus on the palatability of the diet, can also influence feed intake and the number of piglets eating, as reviewed by Middelkoop [70] and Tokach et al. [107]. In addition, it could be speculated that the nutrient density of pre-weaning diets should not be excessive, as a nutrient-dense diet is likely to result in the arrival of excessive amounts of substrate at the end of the ileum, thereby resulting in an overgrowth of potentially pathogenic bacteria, as well as a prolonged feeling of satiety. Although some evidence suggests that a lower density of dietary energy may indeed stimulate post-weaning feed intake [108], this remains to be confirmed for the pre-weaning stage [71,109].

### 3.2. Pre-Weaning Nutritional Strategies to Modulate Gut Health

A variety of nutritional strategies (e.g., highly digestible protein sources, feedstuffs with low fermentability) may also be useful in using creep feed to modulate gut health. Table 3 provides an overview of nutritional interventions using functional amino acids (e.g., glutamine), yeast, prebiotics (e.g., oligofructose), probiotics, synbiotics, fat sources rich in medium-chain fatty acids (MCFA) and various fibre sources (e.g., insoluble and soluble fibres) that have been included in creep feed. Although most of the studies listed here report significant effects on gut morphology, gut microbiota colonisation and maturation of the mucosal immune system, none considers whether the piglets (which had been humanely euthanised in order to measure the gut parameters) had actually ingested the creep feed, and most had studied piglets only until three weeks of age. Nevertheless, these results may suggest that the effects could be even more pronounced for longer creep-feeding periods and higher (individual) levels of creep-feed intake. The effect of the dietary composition of creep feed on post-weaning gut health and performance has yet to be explained. For example, Fouhse et al. [110] report that yeast-derived mannan-rich fraction had positive effects on jejunal morphology at one week post-weaning, but not thereafter (at three weeks post-weaning).

Compared to creep feed, commercial weaner diets have a higher content of non-starch polysaccharide (NSP) and a higher quantity of fibres, amongst other aspects. It would therefore seem useful to formulate creep feed as a means of preparing the gut (digestive system and microbiota) for the weaner diet, rather than as a purely nutritional resource. For example, in a study by van Hees et al. [85], increasing the fermentable (i.e., long-chain arabinoxylan) or non-fermentable fibre (i.e., cellulose) content of milk replacer (fed from Day 2 to Day 13 after birth) and the subsequent creep feed (fed from Day 14 after birth) had the potential to increase the relative weight and SCFA content of the large intestine at weaning on Day 25, as compared to piglets fed with a low-fibre diet. Short-chain fatty acids are degradation products remaining after the fermentation of complex polysaccharides (e.g., arabinoxylan and cellulose). An example of SCFA includes butyrate, which plays an important role in maintaining the gut barrier (i.e., stimulating gut epithelial cell proliferation and degradation). As suggested by this and other studies (Table 3), creep feed can also serve a functional role, in which the nutrient composition of the creep feed may be able to modulate gut health. Additional research is nevertheless needed in order to understand the impact of such strategies on post-weaning growth and gut health [115].

### 3.3. Transition Diet

The success of providing solid feed in the pre-weaning stage with regard to improving performance in the post-weaning stage depends on intake level and the dietary composition of the creep feed and weaner diet, as well as on the similarity amongst these types of feed [79,116]. If there are large differences between creep feed and the post-weaning diet, piglets may not adjust to the post-weaning diet, even despite high creep-feed intake in the pre-weaning stage. This has been demonstrated in a study by Heo et al. [116], in which piglets were given creep feed, a weaner diet or a sow diet pre-weaning. In the post-weaning stage, all of the piglets received the same weaner diet that some litters had already received in the pre-weaning stage. Litters that had been fed creep feed during the pre-weaning stage had the highest total feed intake pre-weaning, and with a similar number of eaters as in the other two treatments. In the first two weeks post-weaning, however, piglets that had received the weaner diet in both the pre-weaning and post-weaning stages had a higher level of post-weaning feed intake than did the other two groups. They also had greater post-weaning body-weight gain than did the creep feed group. Interestingly, the sow-diet group had an intermediate body weight gain in the first two weeks post-weaning, along with improved feed efficiency between Weeks 2 and 5 post-weaning, as compared to the groups receiving the other two dietary treatments. Taken together, these results suggest that the similarity between the composition of pre-weaning and post-weaning diets is more important to post-weaning performance than is the intake level of solid feed in the pre-weaning stage. This suggestion is supported by the results reported by Middelkoop et al. [117], in which two dietary-treatment groups differed substantially in the intake of solid feed during the pre-weaning stage. The difference in feed intake did not persist in the post-weaning stage, however, when the piglets received a diet that differed from the pre-weaning diet (with a similar post-weaning diet across all treatment groups). Moreover, strong positive relationships were found between feed intake pre-weaning and immediate post-weaning, as well as between pre-weaning feed intake and post-weaning piglet growth, when the same diet was given in both the pre-weaning and post-weaning stages [118,119]. It is therefore important to provide the same diet in the pre-weaning period (at least in the last week/days) and the initial post-weaning period, so that piglets will recognise the post-weaning diet in terms of both behaviour (e.g., reduced food neophobia) and physiology.

## 4. Are all Weaned Piglets Alike?

Production losses including growth impairments, increases in the number of veterinary treatments and higher mortality rates are factors that pig producers continue to face during the immediate post-weaning period. The changes that make weaning such a stressful and challenging process for piglets have been described in greater detail in several extended reviews (e.g., Weary et al. [68] and Heo et al. [69]). Briefly stated, they include separation from their dam and littermates, nutritional challenges (e.g., dietary changes from highly digestible sow milk to a less digestible complex solid diet, and changes in feeding habits), mixing with unfamiliar piglets (including establishing a social hierarchy) and adjusting to a new environment.

Piglets vary greatly within weaning batches with regard to their capacity to encounter post-weaning stressors, even though piglets are often fed a common diet at weaning. For example, piglets are weaned at varying weights [7,120], and not all piglets have already eaten solid feed pre-weaning [86,102,121]. Despite common knowledge that weaning weight is crucial to subsequent production performance [120,122], it is often assumed that piglets that are weaned light were also born light [6,123]. Some piglets that are born light are nevertheless able to catch up in growth [7,120]. In addition, recent literature [8,124,125] suggests that various morphometric characteristics suggestive of IUGR (e.g., body mass index, ratio of birth weight to cranial circumference) may explain why some piglets are more likely to end up light at weaning, regardless of birth weight.

The intake of creep feed during the pre-weaning period is one important factor that can influence piglet performance and gut absorption during the post-weaning period [126]. As reported in a study by Bruininx et al. [127], piglets that started eating creep feed pre-weaning had a shorter latency time to eating the post-weaning diet, as compared to non-eaters, as well as a higher feed intake. These results have been supported in other studies [121,128]. Piglets that have consumed creep feed pre-weaning may react differently to the dietary change at weaning than piglets that did not consume creep feed. In a study by Torrallardona et al. [79], piglets were weaned into post-weaning diets with different cereal sources (i.e., barley, rice + wheat bran, corn, naked oats, oats, or rice) and fed creep feed or not during the pre-weaning period. The results suggest that some cereals are more suitable than others, depending on whether a piglet has or has not had access to creep feed. For example, rice + wheat bran resulted in the greatest weight gain and feed intake during the first 21 days post-weaning for piglets that had access to creep feed pre-weaning, whilst piglets that had no access to creep feed performed worst on this cereal source and best on naked oats. No differences were observed at the level of the small intestine that could explain these results. The piglets in the creep-feed group were not classified into eaters and non-eaters [79], however, even though not all piglets are generally considered eaters. These results emphasise the importance of a post-weaning diet that helps both non-eaters and eaters of creep feed to thrive post-weaning.

One possibility for increasing creep-feed intake and the number of eaters could involve increasing the age at weaning [75]. Current changes in legislation with respect to dietary limitations of minerals and the ban on antibiotics, combined with public attitudes concerning farm animal welfare [129], have fuelled a discussion (at least in Europe [130]) about weaning piglets at ages older than 3–4 weeks. Studies on the effect of extending weaning age suggest that delaying weaning from 3 to 4 weeks of age could reduce the post-weaning growth check [131,132], while decreasing post-weaning mortality and faecal pathogenic bacterial counts [132], and improving the intestinal barrier function [133]. Extending weaning age from 4 to 6–7 weeks of age has been suggested to increase feed intake during the immediate post-weaning period [75], while decreasing susceptibility to post-weaning diarrhoea [134]. Taken together, these findings support the proposition that later weaning might result in a more mature immune and gastro-intestinal tract during the immediate post-weaning period. Part of the advantages associated with delayed weaning may be attributed to increased intake of creep feed [75,135], enhanced development of the immune system and the presence of several components in sow milk that can further improve the development of the immune and digestive system [48]. It should nevertheless be noted that weaning later may be beneficial only during the immediate post-weaning period [131,132], and only for piglets that are otherwise weaned light [132,136,137].

In general, two kinds of piglets can be observed at weaning: those that are robust at weaning and thus better able to cope with the associated changes, and those at greater risk of post-weaning diarrhoea. One major reason for the higher probability of post-weaning diarrhoea is anorexia, which results in maldigestion and malabsorption, followed by associated overeating during the subsequent period [22,23]. Especially during the first two weeks post-weaning, a piglet’s digestive system is immature and incapable of digesting and absorbing all nutrients. Overeating can result in excessive undigested nutrients in the gut, which can subsequently provide a substrate for the proliferation of potential harmful bacteria. In this review, the post-weaning period is classified into two physiological phases: an acute phase (from weaning to 5–7 days post-weaning) and a maturation phase (from 5–7 days post-weaning onwards) [138]. This review focusses on feed formulation and feed-processing strategies aimed at supporting the gut health of weaned piglets. These strategies are subsequently combined into nutritional strategies in relation to the acute and maturation phases of weaned piglets.

## 5. Post-Weaning Nutritional Strategies during the Acute Phase

During the acute phase, piglets experience short-term anorexia, with energy intake not being restored to pre-weaning levels until two weeks post-weaning [139]. Combined with the stress of weaning, low feed intake results in impaired gastric barrier and function, intestinal inflammation and histological changes in the small intestine (e.g., villi atrophy), thereby reducing the activity of brush-border enzymes. These changes lead to impairments in the gut mucosal integrity of piglets, thereby increasing the risk of inflammation, resulting in the accumulation of undigested nutrients in the large intestine and an overgrowth of harmful bacteria. The changes ultimately increase the piglets’ susceptibility to enteric bacterial infections [21]. It is therefore important to stimulate feed intake in newly weaned piglets (e.g., by using a transition diet or the same diet that was used in the pre-weaning period). Other nutritional strategies aimed at increasing feed intake post-weaning (e.g., the inclusion of highly palatable ingredients in the post-weaning diet) have been reviewed by others [140,141]. To promote gut health in weaned piglets, a wide range of nutritional interventions have been investigated and reviewed, with several functional ingredients and feed additives receiving the most attention [142,143,144,145]. In the following sub-sections, the nutritional strategies for the acute phase are discussed in relation to their effect on gut health and functionality (i.e., stomach functioning, stomach retention and digestion kinetics, and the health and function of the small and large intestine).

### 5.1. Nutritional Strategies to Support Stomach Functioning

In order to keep piglets healthy, it is important to establish a good gastric barrier and function. The stomach secretes hydrochloric acid (HCl) and enzymes to break down carbohydrates, protein and fats. The secretion of HCl reduces gastric pH and serves two purposes: (1) to facilitate protein digestion and (2) to create a natural barrier. The enzyme pepsinogen that is released is converted into pepsin within an acidic environment, after which protein hydrolysis occurs at pH values of 2.0 to 3.0 [146], while pepsin is inactivated at pH values greater than 5.5 [147]. The ability of the stomach to digest nutrients effectively is dependent on gastric pH and gastric emptying rate, diet composition, meal size and the amount of gastric secretions produced [147]. Young piglets produce only low amounts of HCl, however, resulting in a high stomach pH and, consequently, impaired nutrient digestion [148]. Combined with the infrequent but relatively large amounts of solid feed intake per meal during the immediate post-weaning period [149], impaired nutrient digestion may result in an elevated and variable stomach pH in newly weaned piglets (Figure 2). High stomach pH values increase the amount of undigested protein that enters the intestinal tract, in addition to increasing the risk of post-weaning diarrhoea [69] and possibly impairing the gastric barrier function [150].

It is well known that the buffer capacity of the diet can modulate gastric pH (Figure 3) and the amount of HCl that is needed to acidify the stomach content [147]. The buffer capacity of feed ingredients can be defined according to “acid binding capacity” (ABC), which is defined as the ability of a feed ingredient to resist a change in pH value. The ABC is usually measured as the milliequivalents (meq) of acid or base needed to change the pH of the feed ingredient to the pH end-titration (usually at pH 4) [151]. Feed ingredients with a high ABC-4 value (i.e., acid-binding capacity at pH 4) have a stronger effect on neutralising the pH in the stomach than do feed ingredients with a low ABC-4 value. The ABC-4 value of a diet is apparently dependent on the content of crude protein, ash and minerals [151]. Other factors that might play a role as well include water-holding capacity, intrinsic osmotic pressure [152], the particle size of the diet [147] and the concentration of aspartic and glutamic acid. Amino acids (e.g., glutamic acid and aspartic acid) have a relatively low p*K*_a_ (dissociation constant) value (4.25 and 3.67 respectively; [153]), suggesting that they may contribute to the ability of protein to resist a pH change. The role of feedstuffs, minerals and organic acids on stomach pH is described in the following paragraphs.

#### 5.1.1. The Role of Feedstuffs and Minerals

In general, feedstuffs such as fish meal, milk by-products and vegetable proteins (e.g., rapeseed meal, sunflower seed meal and soybean meal) are known for their high buffer capacity (ABC-4 value of >400 meq/kg), whilst cereals (e.g., wheat, barley and maize) have a low buffer capacity (ABC-4 value of <150 meq/kg; [151]). It could even be suggested that the calcium (Ca) content of starter diets should be limited for a short period (e.g., two weeks), due to its effect on stomach pH (high buffer capacity), thereby favouring stomach functioning without limiting the mineral requirements for bone formation [151,154]. Given that the amount of Ca required for growth is considered less than the amount required for bone ash, González-Vega et al. [155] suggest that a standardised total tract digestible (STTD) Ca content of 0.48% might be sufficient to maximise bone ash content in piglets. The buffer capacity of feedstuffs and minerals should, therefore, be taken into consideration in the formulation of newly weaned piglet diets.

#### 5.1.2. The Role of Organic Acids and Their Interaction with Feed Ingredients

The addition of organic acids to piglet diets can lower the buffer capacity of the diet. Organic acids that can be added to the feed or water can act through (1) their direct bacteriostatic/bactericidal effects and by (2) decreasing stomach pH, thereby improving the onset of protein digestion and reducing the survival of pathogens [156]. Organic acids may also have other modes of action (e.g., as an energy source) along other segments of the digestive tract, as described in previous reviews [157,158]. It should be noted that organic acids can differ in their anti-bacterial properties and their capacity to reduce stomach pH, in which factors including molecular weight, p*K*_a_, chain length and buffer capacity play an important role [159]. Furthermore, the use of a blend of organic acids is preferable to the use of a single acid, as together they have a wider range of anti-bacterial properties [158]. For example, one organic acid may result in the fast reduction in the stomach pH, whilst another may undissociate more quickly at that pH and enter the pathogenic cell wall [160]. It is also important to consider that, due to their pungent odour, some organic acids can have a negative effect on diet palatability when included in high concentrations [159].

The relative success of diet acidification with organic acids on piglet performance, health and nutrient digestibility is also dependent in large part on other feed ingredients [156,161]. This is often overlooked, and it is therefore highlighted briefly in the current review. For example, in diets containing high amounts of milk products, the acidification of the diet with organic acids is less effective. This can be explained by the high lactic acid concentration in the stomach as a result of the fermentation of lactose from milk, which masks the effectiveness of organic acids [161]. Organic acids can be expected to be more effective in diets containing less-digestible feedstuffs (e.g., plant proteins) [156,161]. The effectiveness of the organic acids is also affected by the inclusion rate and particle size of minerals. It is well known that calcium carbonate (provided by limestone) can increase the digesta pH in the proximal part of the gastro-intestinal tract, due to its high acid-binding capacity. Both the inclusion rate and particle size of limestone can increase the surface area, thereby resulting in a faster Ca solubility and release, and thus in a higher buffer capacity of the gastric content [162]. In addition, ZnO has a high buffer capacity [151] and, under conditions in which ZnO can still be added in high amounts to the diet, it can counteract the effectiveness of organic acids.

In summary, knowing the ABC-4 value of each feed ingredient in the diet (e.g., feedstuff, premix, and additives) can help to select ingredients that are suitable for young piglets, and it can help to explain the buffer capacity of the complete diet. Although the reduction in the ABC-4 value of the diet is assumed to lower stomach pH, and although the ABC-4 value is increasingly being used in practise as an important parameter in diet formulations, scientific research is lacking with regard to the evaluation of the optimal ABC-4 (or pH) value of complete piglet diets.

### 5.2. Factors that Prolong Stomach Retention, Promote Gut Health and Modulate Digestion Kinetics along the Small Intestine

The gastric-passage rate is determined by multiple factors, including meal size, stomach mobility and the fraction of the liquids or solids in the stomach, as the liquid fraction passes more rapidly than the solid fraction [163]. Prolonged retention of stomach content can be beneficial for newly weaned piglets, at least to some extent. For example, it can improve protein hydrolysis in the stomach, thereby resulting in greater protein digestibility along the small intestine [164]. Furthermore, the amount of substrate and fermentation end-products (e.g., SCFA) arriving at the end of the ileum can also prolong stomach retention by stimulating the feedback mechanism of hypothalamic satiety signals (Figure 4).

Nutritional strategies that can prolong stomach retention include coarseness or structure and various physiochemical characteristics (e.g., viscosity, water-binding capacity and water-holding capacity) of the feedstuff and diet. Furthermore, these dietary factors can affect the nutrient-digestion kinetics along the small intestinal segments and improve gut health [165]. Although this is an interesting aspect, few studies have been conducted in weaned piglets [164,166]. Current knowledge about nutritional strategies to prolong stomach retention and modulate digestion kinetics along the small intestine in piglets are outlined in the following paragraphs.
Figure 4Interactions between the passage rate of digesta and feed intake. AA = amino acids, CCK = cholecystokinin, GLP-1 = glucagon-like peptide 1, GIP = glucose-dependent insulinotropic polypeptide, HCl = hydrochloric acid, PYY = peptide YY, SCFA = short-chain fatty acids. The figure is adapted from Giger-Reverdin et al. [152] and Lee et al. [167].
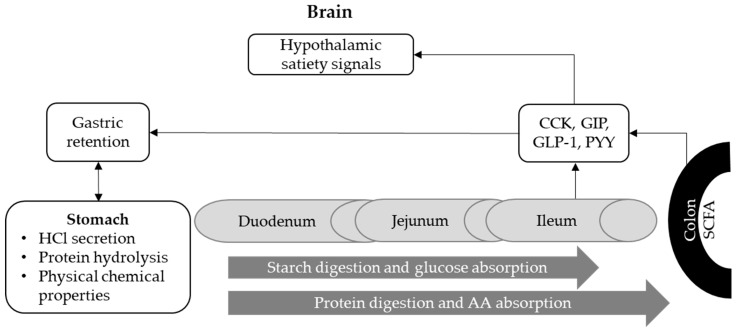


#### 5.2.1. The Role of Particle Size or Feed Structure

Particle size is measured by passing the feedstuff or whole diet through a series of sieves. Different grinding technologies (e.g., roller mill and hammer mill) result in different particle-size distributions. The hammer mill yields finer particles, while the roller mill yields coarser particles, but a more uniform particle size distribution [168,169]. Both fine grinding and pelleting can reduce particle size and improve nutrient digestibility by increasing the surface area for the digestive enzymes. At the same time, however, fine particles can lead to dust and undesired gut problems, including stomach ulcers [170], *Streptococcus suis (S. suis)* colonisation in the stomach [150] and *Salmonella*
*typhimurium* colonisation in the ileum [171]. It is therefore important to have enough structure in the diet by adding coarse particles, in order to ensure good stomach functioning and a gradual transition of particles from the stomach to the gut. As coarse grinding reduces nutrient digestibility, structure may be created by ingredients with low energy values (e.g., wheat bran or wheat straw). It has been demonstrated that feeding piglets coarse rather than finely ground wheat bran (4% inclusion level) during the first two weeks post-weaning is effective in reducing *E. coli* adhesion to the ileal mucosa and in reducing the severity of diarrhoea after an enterotoxigenic *E. coli* challenge [172].

Although studies in weaned piglets on the effectiveness of coarse particles in the diet are limited, studies involving growing pigs have consistently shown beneficial effects. In a study by Hedemann et al. [171], grower pigs (33 kg of body weight) fed a coarse diet, either mash or pelleted (80.1% of the particles were <1000 µm, 15.6% were between 1000 and 2000 µm, 2.1% were between 2000 and 3500 µm, and 2.3% were >3500 µm), exhibited a higher relative empty stomach weight (+7%) than did pigs fed a fine diet (93.6% of the particles were <1000 µm, 6.4% were between 1000 and 2000 µm, 0.0% were between 1000 and 2000 µm, and 0.0% were >3500 µm). Another study, using pigs in the same body-weight class, reports that the pH of the stomach content in the fundus gastric region of pigs that were fed either a coarsely ground diet (geometric mean diameter 671 µm) or a finely ground diet (geometric mean diameter 217 µm) was lower for the coarsely ground diet (pH of 2.5) than for the finely ground diet (pH of 5.0) [150]. If fibre-rich by-products (e.g., wheat bran) are not available, or if they are contaminated with undesired compounds (e.g., mycotoxins), structure can be added to the diet by coarsely grinding a portion (e.g., 5%) of the cereals (e.g., barley).

In addition to the particle size of the feedstuff and/or diet, the structure of feedstuffs may influence gastric emptying rate, as well as digestion kinetics. This has been suggested by Bornhorst et al. [173], who found that the effects of brown rice (which has a higher fibre content and more structure than white rice) were more beneficial than those of white rice on the gastric emptying rate and acidification of the stomach content. More specifically, pigs (20.9 kg of body weight) receiving the brown-rice diet had a slower gastric emptying rate and nutrient transit from the stomach to the small and large intestines than did those that were fed the white-rice diet. This resulted in a lower distal stomach pH for the brown-rice treatment group after ingestion of the meals. The authors suggest that this may have been a result of physical resistance in the stomach due to the accumulation of bran layers from the brown rice [173].

Taken together, these findings suggest that, for young piglets at increased risk of post-weaning disorders, bringing structure to the diet through the use of fibre-rich cereal by-products or the introduction of some cereals in coarse-ground form may support stomach acidification, improve nutrient digestion and reduce the risk of gastro-intestinal infections. Additional research is needed in order to identify the best feed structure (e.g., particles >1.5 mm in pelleted diets) for weaned piglets for optimising gastric function, gut health and nutrient digestibility.

#### 5.2.2. The Role of Heat Treatment

Most pig diets are pelleted or otherwise treated with heat. The conditions of the production process may alter animal health and performance. Extrusion and expansion are “short-time, high-temperature” processes, and are based on a combination of high temperature, moisture, high pressure and shear forces. Expansion is less intensive than extrusion. Intensive feed-production processes may alter the chemical composition of a specific feedstuff or a complete diet [174]. Structures of carbohydrate and protein molecules may be altered during processing. Extrusion can increase starch gelatinisation and increase the solubility of NSP. The increase in the proportion of soluble NSP could affect diet viscosity and water-holding capacity, which could subsequently prolong gastric retention time (as discussed in greater detail below). In addition, extrusion may also improve protein digestion: (1) through the denaturation of proteins, which makes them more accessible to proteolytic enzymes, or by (2) making protein more accessible by breaking down the bonds between starch and proteins that are normally present in cereals [175].

Excessively high processing temperatures can reduce protein digestibility, however, due to Maillard reactions. Furthermore, cereal sources differ in terms of starch-granule size, amylose-to-amylopectin ratios and degree of crystallinity [176]. These differences influence the potential effects of feed processing (e.g., extrusion) on starch gelatinisation and diet viscosity. For example, the heat treatment (e.g., pelleting at different temperatures or expanding followed by pelleting) of diets based on wheat and barley has been shown to increase viscosity, thus yielding the highest diet viscosity at the maximum temperature (Figure 5). No such effect was observed for the maize-based diet.

The effect of heat treatment of barley on viscosity was confirmed by an in vivo study (using extrusion) conducted by Rodrigues et al. [166] in weaned piglets. In that study, extruded barley resulted in increased viscosity of digesta at the proximal small intestine (from 1.95 to 5.09 centipoise (cps)) and distal ileum (from 3.14 to 6.56 cps), as compared to raw barley. No such effect was observed for rice [166]. Recent studies [166,178] have generated new insight into the effect of extrusion on the kinetics of starch digestion in pigs. In general, the starch digestion appears to be faster for extruded cereals than it is for ground cereals, as measured along the small intestine (Table 4). Further investigation is needed with regard to the potentially beneficial effects of large particles in coarsely ground cereals (as discussed above), heat-processed cereals or a combination thereof on gastric retention and digestion kinetics in weaned piglets.

#### 5.2.3. The Role of Physicochemical Characteristics

Feed ingredients have a variety of physicochemical properties, including viscosity and hydration (i.e., water-holding capacity and water-binding capacity). Soluble (fermentable) fibres are believed to increase the viscosity and water-holding capacity (WHC) of the digesta content of the stomach and small intestine [179]. Studies on the effect of soluble (usually fermentable) and insoluble (usually inert) fibre on stomach retention and passage rate have yielded inconsistent results. For example, Fledderus et al. [164] report a 10% reduction in the gastric-emptying rate due to the addition of 1% dietary carboxymethylcellulose, along with a nearly seven-fold increase in diet viscosity for piglets at 3 weeks post-weaning. Gastric protein hydrolysis and ileal crude protein digestibility increased as well. The authors further report that supplementation with both guar gum and cellulose (i.e., insoluble fibres) increased the gastric passage rate and the viscosity of the ileal digesta, although only guar gum increased the total tract-retention time [180]. Another study suggests a linear increase in digesta viscosity with increasing intake of soluble fibre [179], although an increase in viscosity in the small intestine could increase the risk for *E. coli* proliferation [181].

Both WHC and water-binding capacity (WBC) are defined as the capacity to retain water. The difference is that WBC is defined after centrifugation, whereas WHC is not. Fibre sources that are rich in pectin (e.g., pea cotyledon and sugar beet pulp) have exhibited greater WBC (7.6 and 8.7 kg/kg dry matter) than have been observed for hull fractions (3.7 kg/kg dry matter) [182,183]. Water-binding capacity has been suggested to have a negative influence on feed intake when included at extreme levels in the diet [184]. Insoluble fibres have been suggested to stimulate feed intake by increasing the gastric passage rate. If the WBC content or viscosity of the stomach digesta is too high, however, feed intake will be depressed by gastric distension [185]. In one study, Ndou et al. [186] explore the role that the WHC of bulk feed plays in predicting the feed intake of piglets weighing 18 kg. The authors report that a WHC of >4.5 g water/g dry matter resulted in lower feed intake, due to prolonged stomach retention. In general, a certain level of diet viscosity and WHC is beneficial for prolonging gastric retention time, although excessive levels can reduce feed intake.

### 5.3. The Role of Main Nutrients in Supporting the Health and Function of the Gastro-Intestinal Tract

#### 5.3.1. The Role of Crude Protein Level, Quality and Functional Amino Acids

Diets for weaned piglets have traditionally been quite luxurious and rich in energy and crude protein (CP), in order to compensate for the low feed intake that occurs during the immediate post-weaning phase. At the same time, however, high CP levels are known to be a major risk factor for post-weaning diarrhoea in weaned piglets [142,187,188]. The fermentation of undigested CP in the end of the small intestine and the colon may lead to the proliferation of pathogenic bacteria. In addition, fermentation creates harmful products for the intestine (e.g., biogenic amines, ammonia and other toxins), which can result in diarrhoea. Meanwhile, the excess of amino acids (AA) cause energy to be deaminated and excreted as urea through urine [171]. For these reasons, the CP levels of piglet diets should be lowered at least during the first two weeks post-weaning.

The selected CP sources in post-weaning piglet diets should be easily digestible and have minimal impact on stomach pH, and the amount of CP that may be attached to fibre parts should be limited. The literature on interactions between fibre and protein fermentation has been reviewed by Jha and Berrocoso [189]. In short, given the limited fermentation capacity of weaned piglets, excessive levels of fermentable fibres can increase microbial protein in the large intestine. In addition, protein is generally digested and absorbed in the small intestine, whereas fibre is fermented by microbiota in the large intestine. If too much CP is attached to fibre, this will reduce CP digestion and absorption in the small intestine, possibly increasing the risk of post-weaning diarrhoea.

Another factor that should be considered is the rate of digestion for CP-rich sources. In a study by Montoya et al. [190], the rate of digested protein entering the small intestine strongly predicted the disappearance of AA in the small intestine. The digestion and absorption of protein occur primarily in the first half of the small intestine, while poorly digestible proteins entering the small intestine are digested and absorbed throughout the entire small intestine. Chen [191] reports differences in CP digestibility along the small intestine in pigs. The results suggest that dried plasma protein is highly digestible, since CP digestibility was 59% in the proximal small intestine, as compared to soybean meal, which yielded poor CP digestion (26%) in the proximal small intestine. Similar digestibility coefficients were reached only at the end of the ileum (74% for soybean meal) compared with 76% at ¾ of the small intestine for dried plasma protein. These results suggest that, when determining the digestibility of a CP-rich source, the distal ileum might not represent what actually happens in the piglet. Future digestibility research should focus more on the kinetics of CP digestibility, instead of considering only the distal ileum.

Another factor that could influence the digestibility of CP-rich feedstuffs is the grinding diameter, with fine grinding increasing the surface area for the digestive enzymes. Although results with respect to particle-size reduction and digestibility in CP ingredients have been inconsistent, some authors suggest that fine grinding improves nutrient digestibility and the feed-conversion ratio, while others do not [192,193,194]. In addition, the effectivity of fine grinding for a given ingredient may be dependent on the structure of the complete diet. As mentioned earlier, coarser diets influence stomach function (e.g., by increasing stomach retention) and can improve gut morphology [171]. It is therefore plausible to speculate that longer retention time due to the combination of a coarser diet and fine-ground protein sources would result in an additive effect with respect to protein hydrolysis in the stomach and digestion in the small intestine. For example, the expected effects of a coarse diet on gastric pH, solid gastric content and nutrient flow from the stomach to the small intestine may help to improve the digestion and absorption of fine CP-rich feedstuffs.

In order to avoid AA deficiencies, it is important to keep AA in balance when reducing the CP content, since some AA become limiting. Piglets need AA in order to deposit protein and renew the turnover of the body protein [195]. Lysine is typically the first limiting AA for pigs fed with cereal-based diets, and its major function is to maintain body-protein synthesis [196]. At the same time, however, other essential and semi-essential AA may be utilised in post-weaning piglet diets, given their role in promoting intestinal development and health (e.g., improving intestinal morphology, increasing the proliferation of epithelial cells, and maintaining intestinal mucosal integrity). Examples include glutamine, threonine and tryptophan (reviewed by Mou et al. [197]).

#### 5.3.2. The Role of Dietary Fibres

The inclusion of dietary fibre in post-weaning diets has been controversial, as fibre can reduce feed intake and nutrient digestibility, thus increasing the risk of proliferation of pathogenic bacteria in the gastro-intestinal tract. These controversial results are partly due to the lack of information regarding the functional effects of dietary fibre, including the modification of the physicochemical characteristics of the digesta or the fermentation characteristics of various feedstuffs. The beneficial effects of fibre in weaned piglets have been intensively reviewed by Molist et al. [198] and by Flis et al. [199]. In addition to the aforementioned influence of physicochemical properties on stomach retention and gut viscosity, it could also be interesting to rank fibre according to its fermentation characteristics: (1) inert fibre (ICHO), consisting of carbohydrates that are not digested and fermented in the gastro-intestinal tract of piglets; (2) fermentable fibre (FCHO), consisting of carbohydrates that are not digested, but are fermented in the large intestine of piglets. The main ingredients that are generally used as sources of ICHO are wheat bran, oat hulls, sunflower hulls and wheat straw. The main sources of FCHO are sugar beet pulp, citrus pulp, chicory pulp and inulin.

In a review article, Flis et al. [199] conclude that the use of different sources of inert fibre to dilute diets during the immediate post-weaning period increases feed intake by between 4% and 54%, as compared to a control diet low in inert fibre. The same authors also report that the supplementation of post-weaning diets with different sources of FCHO reduces feed intake by between 7% and 28%, as compared to a control diet low in inert fibre [199]. Similar results are reported by Montagne et al. [200] in a study on interactions between fibre fermentability in the diet and the sanitary conditions of the farm with regard to the growth and health status of weaned pigs. In this study, piglets fed with a high level of FCHO (6% sugar beet pulp and 2% soybean hulls/kg) and allocated in rooms with high infection pressure exhibited lower feed intake during the first two weeks post-weaning, as well as a higher incidence of diarrhoea than did piglets from the other experimental groups. The authors conclude that the inclusion of ingredients with FCHO immediately post-weaning imposes an additional risk factor for piglet health and growth, especially under poor sanitary conditions. In contrast, Gerritsen et al. [201] observed that diluting the diet with 12% ICHO during the first two weeks post-weaning resulted in higher feed intake, stomach weight and amylase activity of the brush-border enzymes in the ileum, as compared to piglets fed with the control diet (low in ICHO). Moreover, the inclusion of ICHO resulted in lower *E. coli* counts in the ileum and colon digesta. The inclusion of fibre sources rich in ICHO in post-weaning diets can therefore be used to dilute the dietary energy level and to increase feed intake and the passage rate of the intestinal content, while reducing the proliferation of pathogenic bacteria in the small intestine. Based on the ICHO and FCHO levels used by Gerritsen et al. [201], it could be speculated that the ratio of FCHO to ICHO during the immediate post-weaning period should be <1.

In conclusion, with regard to dietary fibre, it is advisable to include moderate levels of ICHO in the diet in order to: (1) dilute the diet and avoid diarrhoea due to the accumulation of undigested nutrients; (2) to help piglets to increase stomach capacity and restore the activity of brush-border enzymes.

#### 5.3.3. The Role of Fat

Another important dietary factor during the immediate post-weaning period is the fat source used. In addition to its necessity as an energy source, fat enhances the quality of pellets. While many studies have examined the role of dietary fibre and CP content in the post-weaning piglet diet, the role of fat in piglet nutrition has received only limited attention. This gap in knowledge is surprising, given that the ability of piglets to digest fat is impaired during the immediate post-weaning period, especially in case of diarrhoea [202]. The fat source used in post-weaning piglet diets should therefore be easily digestible. Fat digestion requires the emulsion of conjugated bile acids with fats, followed by the transport and absorption of the fat micelle. The digestibility of dietary fat is directly related to the ability to form micelles, which is associated with the length and double bonds of specific fatty acids. For example, in contrast to long-chain fatty acids, SCFA can easily form micelles, in addition to diffusing directly into enterocytes [203]. It is therefore not surprising that shorter chain length is associated with better fat digestibility [203]. In addition, unsaturated fatty acids are regarded as being more easily digestible than are saturated fatty acids. The ratio of unsaturated fatty acids to saturated fatty acids (i.e., the “US ratio”) should, therefore, be considered in order to ensure proper digestion of the fat used in post-weaning piglet diets. A higher US ratio is required for young piglets than for older pigs [203].

Medium-chain fatty acids (e.g., coconut fatty acids, palm kernel oil), which contain 6–12 carbon atoms, are also regarded as a readily available energy source, as they are easily digestible (i.e., they can be absorbed intact into the intestinal epithelial enterocytes). They may also have certain antimicrobial [204] and immunomodulatory effects, in addition to providing a good source of energy for gastro-intestinal cells (as reviewed by Jackman et al. [205]). It has been suggested that MCFAs disrupt the phospholipid membrane of bacteria and that they can have a bacteriostatic (inhibiting the growth of bacteria) or bactericidal (directly killing bacteria) effect, especially for Gram-positive bacteria and lipid bilayer viruses [205]. It is nevertheless important to consider the fact that different MCFAs can target different pathogens, with varying effectivity [205], with MCFA being the preferred fat source in post-weaning piglet diets.

Omega-3 fatty acids (ω-3) are regarded as immune-modulating nutrients, due to their anti-inflammatory properties. It has been suggested that ω-3 could play a role in attenuating the intestinal inflammation that is normally observed at weaning, with the ratio ω-6:ω-3 being an important parameter to consider. For example, lower plasma concentrations of tumour necrosis factor-α (TNF-α), the principle mediator of inflammation, were found for piglets fed with a diet supplemented with ω-3 fatty acids, as compared to piglets fed with the control diet [206]. Similarly, Huber et al. [207] replaced maize oil with fish oil at 1.25%, 2.5% and 5% in order to achieve ω-6:ω-3 ratios of 5:1, 3:1 and 1:1, respectively. They report that the acute phase protein (APP) haptoglobin decreased with increasing fish-oil supplementation after challenging weaned piglets with ovalbumin and lipopolysaccharide (LPS). This indirectly suggests reduced production of pro-inflammatory cytokines, and thereby reduced production of APP (e.g., haptoglobin) by the liver [207]. Moreover, Shin et al. [208] report that reducing the ω-6:ω-3 ratio to 4:1 post-weaning suppressed the inflammatory response, as evidenced in lower IL-1β and PGE2 levels, amongst other indicators. Future diet formulations should therefore devote greater attention to optimising the ω-6:ω-3 ratio for piglet diets during the acute post-weaning phase.

## 6. Post-Weaning Nutritional Strategies during the Maturation Phase

During the maturation phase, gut integrity and function are restored, due to higher feed intake and strengthened digestion, absorption and fermentation capacity, as reviewed by [209,210,211]. In this phase, it is important to maintain piglet health while improving growth and preparing for the growing–finishing phase. Similar to the acute phase, it remains important to have structure in the diet and a low ABC during the maturation phase, in order to support a good gastric barrier, thereby promoting digestibility and gut health. There are also several differences between the nutritional approaches needed in the acute and maturation phases. As mentioned before, during the maturation phase, piglets recover their feed intake as their gastro-intestinal tracts mature. For example, nutrient digestibility increases with age, with the protein digestibility of wheat-protein-based diets increasing over time from 47% (Day 7), to 70% (Day 14), 76% (Day 21) and 78% (Day 28) [212]. During the maturation phase, diets can be formulated to focus on increasing lysine intake while maintaining a balanced AA profile to improve piglet performance. This can be achieved by increasing the ratio of standardised ileal digestible (SID) lysine to energy in the piglet diet. Testing a range of SID lysine levels from 10.3 to 15.1 g/kg in piglets with body weights of 7–16 kg, the SID lysine requirement for growth ranged from 12.9 to 13.4 g/kg, using linear and quadratic broken lines, respectively, equivalent to 16.8 g of SID per kg of gain [213]. For piglets of 10–25 kg, Kendall et al. [214] identify a requirement of 19 g SID lysine per kg of gain. In another study (Table 5), growth performance and feed efficiency exhibited a clear positive response to increases in the level of SID lysine. For the reasons discussed above, however, this high level of SID lysine should ideally be achieved by using synthetic AA, rather than by increasing the CP content of the diet. It should be noted that restrictions relating to the levels of Cu and Zn in piglet diets can have detrimental effects on piglet performance. As reported by Bikker et al. [215], decreasing Cu from 170 to 100 mg between 28 and 40 days post-weaning could result in a reduction of approximately 600 g in body weight at the end of the post-weaning period (25.2 vs. 24.6 kg body weight at the age of 68 days). These results highlight the importance of increasing the SID Lys/NE in the last 3–4 weeks of the post-weaning period in order to counteract this loss in performance.

During the maturation phase, piglets also have an increased fermentation capacity, and they are able to generate energy from the absorption of SCFA [216]. This allows nutritionists to increase FCHO levels in this phase. In a recent review, Jha and Berrocoso [189] indicate that feeds with high contents of FCHO (e.g., citrus pulp and sugar beet pulp) are more effective as fibre sources for reducing nitrogen loss in manure for piglets heavier than 15 kg of body weight, as compared with ICHO. These results suggest that the ratio of FCHO to ICHO should be >1.5 during the maturation phase. Hermes et al. [217] report decreased faecal *E. coli* counts (7.77 vs. 6.86 log_10_ of CFU/g of faeces) and increased ratio of lactobacilli to enterobacteria (0.76 vs. 1.37) in the faeces of pigs receiving diets supplemented with a combination of 4% wheat bran and 2% sugar beet pulp/kg, as compared to a low-fibre diet. The authors attribute the beneficial effects of the high-fibre diet to changes in the microbial profile within the gastro-intestinal tract, due to the increased production of SCFA from the fermentation of dietary pectins. Overeating should also be avoided during the maturation phase, as it could increase the risk of pathogenic bacterial infections (e.g., *S. suis*) [69]. As reported by Correa-Fiz et al. [218], supplementation of post-weaning diets with MCFA and a supplement containing MCFA and natural plant extracts (with anti-inflammatory properties) reduced clinical signs (e.g., meningitis) associated with *S. suis* on Day 43 post-weaning. Considering that most *S. suis* infections occur around 3 weeks post-weaning, it might be beneficial to supply MCFA for its antibacterial effects, in addition to supplying long-chain fatty acids (e.g., vegetable oils) as energy sources. Compared to the acute phase, the US ratio of the diet can be reduced during the maturation phase, as piglets are expected to have greater fat-digestibility capacity [219].

Taken together, in response to the physiological changes that piglets undergo during the acute and maturation phases, several post-weaning nutritional strategies for piglets have been formulated, as summarised in Scheme 1.

## 7. Conclusions

As demonstrated by this review of the relevant literature, early-life nutritional strategies can modulate gut health, and thereby piglet performance, in both the short and longer term. Given that the critical period for gut (microbiota and immune) development occurs before weaning, further research is needed in order to identify how functional ingredients in supplemental milk and creep feed can be used to prepare piglets for weaning. The findings from this review highlight a lack of information regarding the effects of pre-weaning nutritional strategies on post-weaning gut health and performance. In addition, the composition of and interaction between pre-weaning and post-weaning diets apparently play a very important role in their potential to support piglet gut health and performance around weaning. It is therefore crucial to adopt a structured approach to nutritional strategies from the first days after birth to the first weeks post-weaning, in order to reduce gastro-intestinal problems in piglets, thereby decreasing the associated morbidity, mortality and antimicrobial use. In the first days of life, the goal of dietary modulation is to increase the survival of piglets. In the weeks following birth, early-life nutritional programming should focus on stimulating gut development and maturation. As weaning approaches, the focus should shift to stimulating the intake of solid feed, in order to prepare piglets for weaning, as well as to preventing undereating and overeating, and the associated post-weaning growth dip and gastro-intestinal dysbiosis. Given that weaning is highly stressful for piglets, nutritional strategies should first target gut health and functionality. The results reported in the current review can be combined to formulate a comprehensive dietary approach that supports stomach function and optimises the kinetics of nutrient digestion along the small intestine, in order to minimise undigested substrate and the risk of over-proliferation of bacteria at the end of the small intestine. Optimising this approach for newly weaned piglets will require future research in order to enhance understanding with regard to digestion kinetics occurring between nutrients in the gut and the role of feed processing on the modulation of stomach functioning. From about one-week post-weaning onwards, the focus of nutritional strategies can shift gradually towards promoting piglet growth performance, in order to improve body weight at the end of the post-weaning period and to prepare piglets for the growing period, while maintaining gut health. Current pig production will continue to face challenges relating to gut health and performance in piglets, particularly in light of expected further restrictions regarding Cu and Zn. This emphasises the need to study the interplay between nutrition and gut health.

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
