# Peer review of "Using Nutritional Strategies to Shape the Gastro-Intestinal Tracts of Suckling and Weaned Piglets"

_animals, 2021, doi:10.3390/ani11020402_

Round 1
Reviewer 1 Report
Dear authors,
my congratulations on the thorough work you have done.
Below you can find some corrections and finally some suggestions.
line n. 47 - "the latter has" please replace with "the litter has"
line n. 64 and following - if there are two subjects (reduction of colostrum and milk intake), I think the verbs should be in the plural form, don't you?
line n. 418 - Is the espression "reduece food neophobia" correct also for animals?
line n. 492 - repetition: "acute phase phase"
line n. 520 and following - "HCL" please replace with "HCl"
line n. 689 - Bornhorst et al. is not reference ⌈152⌉.
line n. 722 - "maillard" please replace with "Maillard"
line n. 763 - I think you meant "carboximethyl cellulose"
I would like to congratulate the authors on their comprehensive and thorough work, which takes into account the different aspects of managing the development of the piglet's digestive system in the pre- and post-weaning phase.
Because of its breadth, the work requires to summarise the various opportunities provided by research on nutritional strategies. This is done in Scheme I, which is in my opinion too much succinct/schematic.
I would suggest that, for example, some indications on effectiveness or degree of applicability of the technology might help to transfer your work to practice
The conclusions could also provide some input for future research, in addition to the indications of feed strategies to be investigate, that are nevertheless identified.
Author Response
Reviewer #1
I don't feel qualified to judge about the English language and style
The English language and style have been checked by Wageningen in'to Languages. Their changes made to the document are tracked (see document Animals_1090251_Tracked_Changes). As there were a substantial number of changes these are not reported in this rebuttal, but only tracked in the document.
Changes made to the document as suggested by the reviewers are marked yellow in the document Animals_1090251_Tracked_Changes. Track changes have been used for changes regarding to English language and style.
Dear authors,
my congratulations on the thorough work you have done.
Below you can find some corrections and finally some suggestions.
Thank you for your kind words and suggestions. Please find our responses to the suggestions below.
line n. 47 - "the latter has" please replace with "the litter has"
We agree that the latter can be clarified by mentioning litter size. We therefore replaced “The latter has increased” by “Litter size has thereby increased”.
line n. 64 and following - if there are two subjects (reduction of colostrum and milk intake), I think the verbs should be in the plural form, don't you?
We agree with the reviewer and have replaced “Overall, the reduced colostrum and milk intake increases the risk of malnutrition or even starvation, increases the risk of hypothermia and disease susceptibility, and results in variable growth rates within batches” by “Taken together, reductions in the intake of colostrum and milk increase the risk of malnutrition or even starvation, as well as the risk of hypothermia and disease susceptibility, ultimately resulting in variable growth rates within batches”.
line n. 418 - Is the espression "reduece food neophobia" correct also for animals?
This is a valid question, as generally feed is used for animals rather than food. However, when discussing food neophobia in animals, the term food neophobia instead of feed neophobia has been used consistently. We will therefore keep the term food neophobia. Some example references in dairy calves, piglets and rats:
J.H.C. Costa, R.R. Daros, M.A.G. Von Keyserlingk, D.M. Weary. Complex social housing reduces food neophobia in dairy calves. J. Dairy Sci., 97 (2014), pp. 7804-7810, 10.3168/jds.2014-8392
- Oostindjer, J. Munoz, H. Van den Brand, B. Kemp, J.E. BolhuisMaternal presence and environmental enrichment affect food neophobia of piglets. Biol. Lett., 7 (2011), pp. 19-22, 10.1098/rsbl.2010.0430
- Modlinska, R. Stryjek. Food neophobia in wild rats (Rattus norvegicus) inhabiting a changeable environment - a field study. PLoS One, 11 (2016), p. e0156741, 10.1371/journal.pone.0156741
line n. 492 - repetition: "acute phase phase"
Thank you for noticing this type error. We have corrected it.
line n. 520 and following - "HCL" please replace with "HCl"
Thank you for noticing this type error. We have replaced HCL by HCl throughout the manuscript and in Figure 4.
line n. 689 - Bornhorst et al. is not reference ⌈152⌉.
Thank you for noticing this editing error. We have checked this and [152] was added manually instead of using a reference manager, resulting in referring to the wrong study in the reference list. Bornhorst et al. was already in the reference list, so correcting the citation number was sufficient.
line n. 722 - "maillard" please replace with "Maillard"
Thank you for noticing this error. We have corrected it.
line n. 763 - I think you meant "carboximethyl cellulose"
Thank you for noticing this error. We have replaced “carboxymehyl cellulose” by “carboxymethylcellulose”. The latter is how it is written by the reference.
I would like to congratulate the authors on their comprehensive and thorough work, which takes into account the different aspects of managing the development of the piglet's digestive system in the pre- and post-weaning phase.
Thank you for your kind words.
Because of its breadth, the work requires to summarise the various opportunities provided by research on nutritional strategies. This is done in Scheme I, which is in my opinion too much succinct/schematic. I would suggest that, for example, some indications on effectiveness or degree of applicability of the technology might help to transfer your work to practice.
We agree with the reviewer that adding indications of effectiveness of the strategies or the status of applicability may help to transfer the review into practise. However, we have decided not to include this as this is dependent on the diet composition that is used and farm management. Diet composition and farm management are very different between countries, even within Europe. It is therefore very hard to compare the effectiveness of strategies that have been tested under different conditions. Recommendations are therefore dependent on the individual situation. In addition, existing feed ingredients are changing rapidly over time with respect to quality and price and new (more sustainable) feed ingredients becoming available on the market.
The conclusions could also provide some input for future research, in addition to the indications of feed strategies to be investigate, that are nevertheless identified.
Thank you for this comment. We have indeed provided indications for future research in the conclusion section.
Reviewer 2 Report
The authors have conducted a very thorough and interesting review of shaping the gastro-intestinal tract of suckling and weaned piglets using nutritional strategies.
One thing that is missing as it is a review article is what key words were used for the review and what systematic approach? As it is quite an extensive review then maybe this should come under supplementary or between the introduction and chapter 2 (around Line 107). The following search terms have been used XXXX in scholar? Pubmed?
There is also something wrong with the page numbering (line numbers are correct).
In the first section the newer findings of Amdi and Lynegaard could also be added. So I suggest to include a recent paper showing that IUGR piglets do have differences in immunology compared to normal piglets at weaning:
Intrauterine growth restriction in piglets alters blood cell counts and impairs cytokine responses in peripheral mononuclear cells 24 days post-partum | Scientific Reports (nature.com)
And another one on stomach capacity
Animals | Free Full-Text | The Stomach Capacity is Reduced in Intrauterine Growth Restricted Piglets Compared to Normal Piglets (mdpi.com)
And also in line 60
Intrauterine growth restricted piglets defined by their head shape ingest insufficient amounts of colostrum1 | Journal of Animal Science | Oxford Academic (oup.com)
Line 93: suggest changing “to that end” to “Therefore,…”
Line 110 does not read quite right; suggest: “…than the sow has productive teats, as the sow cannot rear them on her own” I guess if the sow has 25 - 30 piglets there is no way she can rear them on her own!
Line 289 – suggest: increase of individual dry matter intake
Line 441 – Here Lynegaard 2020 could be added:
Body composition and organ development of intra-uterine growth restricted pigs at weaning | animal | Cambridge Core
Line 550 – suggest to write reviewed by Heo and colleagues (65).
Line 798 – Lynegaard 2021
Suggest to add a recent article confirming that a diet with low CP levels from weaning to about 15 kg BW had a reducing effect on diarrhea, but decreased ADG without affecting the FCR.
Low protein diets without medicinal zinc oxide for weaned pigs reduced diarrhoea treatments and average daily gain - ScienceDirect
Line 809 – comma after also and again in line 825. Generally I think there might be missing some commas throughout.
Author Response
Reviewer #2
English language and style are fine/minor spell check required.
The English language and style have been checked by Wageningen in'to Languages. Their changes made to the document are tracked (see document Animals_1090251_Tracked_Changes). As there were a substantial number of changes these are not reported in this rebuttal, but only tracked in the document.
Changes made to the document as suggested by the reviewers are marked yellow in the document Animals_1090251_Tracked_Changes. Track changes have been used for changes regarding to English language and style.
The authors have conducted a very thorough and interesting review of shaping the gastro-intestinal tract of suckling and weaned piglets using nutritional strategies.
One thing that is missing as it is a review article is what key words were used for the review and what systematic approach? As it is quite an extensive review then maybe this should come under supplementary or between the introduction and chapter 2 (around Line 107). The following search terms have been used XXXX in scholar? Pubmed?
This review in its whole cannot be classified as a systematic review. Therefore the search terms are not included. It is possible to do a systematic approach to create Table 1-3 if preferred by the reviewer.
There is also something wrong with the page numbering (line numbers are correct).
Thank you for noticing this. Some pages are horizontal, because of some larger tables, and there are therefore multiple sections in the text. At a new section the page numbering started at 1 at those new sections. This has been corrected, so the page numbers are now continuing over the sections.
In the first section the newer findings of Amdi and Lynegaard could also be added. So I suggest to include a recent paper showing that IUGR piglets do have differences in immunology compared to normal piglets at weaning: Intrauterine growth restriction in piglets alters blood cell counts and impairs cytokine responses in peripheral mononuclear cells 24 days post-partum | Scientific Reports (nature.com). And another one on stomach capacity. Animals | Free Full-Text | The Stomach Capacity is Reduced in Intrauterine Growth Restricted Piglets Compared to Normal Piglets (mdpi.com).
And also in line 60. Intrauterine growth restricted piglets defined by their head shape ingest insufficient amounts of colostrum1 | Journal of Animal Science | Oxford Academic (oup.com).
We agree to add those references. We have added Amdi et al. 2020 to this part of the sentence “and have a higher disease susceptibility [1,8, Amdi et al., 2020]”. We have added Amdi et al. 2013 and Lynegaard et al. 2020 by adding a new sentence to the manuscript: “On top, IUGR piglets consume less colostrum than piglets with a normal head morphology [Amdi et al., 2013] and also have a lower stomach capacity to do so [Lynegaard et al., 2020].”
Line 93: suggest changing “to that end” to “Therefore,…”
Thank you for this suggestion. We changed “To that end” to “In response to the developments outlined above”, which was suggested by the native speaker from Wageningen in’to Languages.
Line 110 does not read quite right; suggest: “…than the sow has productive teats, as the sow cannot rear them on her own” I guess if the sow has 25 - 30 piglets there is no way she can rear them on her own!
We agree with the reviewer. We have replaced “..as the sow may not be able to rear them on her own” by “as the sow cannot rear them on her own”.
Line 289 – suggest: increase of individual dry matter intake
Thank you for this suggestion. We have checked the section titles and clarified them as following. We have changed “3.1. Increase individual dry matter intake” into “3.1. Pre-weaning nutritional strategies to stimulate individual dry-matter intake”. In addition, we have replaced “3.2. Modulate gut health” into “3.2. Pre-weaning nutritional strategies to modulate gut health”.
Line 441 – Here Lynegaard 2020 could be added: Body composition and organ development of intra-uterine growth restricted pigs at weaning | animal | Cambridge Core
Thank you for this suggestion. We have added the reference to the following sentence: “In addition, recent literature [122,123] suggests that various morphometric characteristics suggestive of IUGR (e.g. body mass index, birth weight to cranial circumferences) may explain why a piglet is more likely to end up light at weaning irrespective of birth weight.”
Line 550 – suggest to write reviewed by Heo and colleagues (65).
We found out that the reference behind this sentence was not the correct reference. We have changed the reference by two correct references. This suggestion of the reviewer is therefore no longer applicable.
Line 798 – Lynegaard 2021. Suggest to add a recent article confirming that a diet with low CP levels from weaning to about 15 kg BW had a reducing effect on diarrhea, but decreased ADG without affecting the FCR. Low protein diets without medicinal zinc oxide for weaned pigs reduced diarrhoea treatments and average daily gain – ScienceDirect.
Thank you for this suggestion. We have added Lynegaard et al 2021.
Line 809 – comma after also and again in line 825. Generally I think there might be missing some commas throughout.
The English language and style have been checked by Wageningen in'to Languages. They replaced the word “also” when it was used at the start of the sentence. This suggestion by the reviewer is therefore no longer applicable. They have also checked the commas throughout the manuscript. Their changes made to the document are tracked (see document Animals_1090251_Tracked_Changes). As there were a substantial number of changes these are not reported in this rebuttal, but only tracked in the document.
Reviewer 3 Report
Huting et al. have reported in this extensive review various strategies on shaping the GIT of suckling and weaned piglets. This extremely comprehensive review has managed to cover a multitude of factors that have a significant impact on the health and development of piglets from birth to maturation. Furthermore, this review also takes into account the effects of these nutritional strategies on the gut health and gut microbiome. There is growing knowledge on the gut microbiome and this area of research is getting more focus in numerous nutritional studies. Therefore, it is becoming a very important part that can be modulated and "programmed" in order to increase gut health and decrease animal mortality. This review article covers multiple topics of interest for the field of pig production and is relevant for the field. The manuscript text is very well written, clear and easy to follow. Congratulations on the extensive work and details provided in this paper.
Author Response
Reviewer #3
Huting et al. have reported in this extensive review various strategies on shaping the GIT of suckling and weaned piglets. This extremely comprehensive review has managed to cover a multitude of factors that have a significant impact on the health and development of piglets from birth to maturation. Furthermore, this review also takes into account the effects of these nutritional strategies on the gut health and gut microbiome. There is growing knowledge on the gut microbiome and this area of research is getting more focus in numerous nutritional studies. Therefore, it is becoming a very important part that can be modulated and "programmed" in order to increase gut health and decrease animal mortality. This review article covers multiple topics of interest for the field of pig production and is relevant for the field. The manuscript text is very well written, clear and easy to follow. Congratulations on the extensive work and details provided in this paper.
Thank you very much for your kind words.